# The Relationship between Parent–Child Attachment, Belief in a Just World, School Climate and Cyberbullying: A Moderated Mediation

**DOI:** 10.3390/ijerph19106207

**Published:** 2022-05-19

**Authors:** Shengnan Li, Xiaoxian Wang, Yangang Nie

**Affiliations:** Department of Psychology, School of Education, Guangzhou University, Guangzhou Higher Education Mega Center, No. 230, Outer Ring West Road, Panyu District, Guangzhou 510006, China; lishengnan010@gmail.com (S.L.); wxx90350124@163.com (X.W.)

**Keywords:** parental attachment, cyberbullying, just world belief, school climate

## Abstract

The present study investigated the relationship between parental attachment and cyberbullying, with the just world belief of the mediator and school climate being the moderator. We collected survey data from 750 middle school students and analyzed the data through mediation and moderation models. The results indicated that after controlling for gender and age, parent–child attachment was negatively related to cyberbullying, with a just world belief significantly mediating this relationship. What is more, school climate moderated the second half of this relationship, as we predicted. We offered possible reasons for the results. Limitations and direction for future studies were discussed.

## 1. Introduction

With the advancement of technologies and the increasing popularity of various electronic devices, a new form of bullying, cyberbullying, has emerged in the public eye due to daily access to social media outlets [1]. It refers to “harm inflicted through the medium of electronic text” [2]; however, it can also occur through other means, such as video [3]. For victims of cyberbullying, the adverse psychological effects they may experience include increased likelihood of suicide, high levels of anxiety, depression and stress [4], and increased severity of depression [5].

Ecological systems theory [6] holds that individuals develop in a multi-layered system that includes microsystems, mesosystems, exosystems, macrosystems and chronosystem. The systems that are most directly relevant to individuals are microsystems and mesosystems. The former often includes the immediate environment that one stays in, such as family and school, whereas the latter refers to the interconnectedness between all microsystems. For instance, a child’s interpersonal style in school is usually affected by how they are interacting with their caregivers at home. Based on this theory, it is essential to consider not only individual factors, but also familial and environmental factors when it comes to individual behaviors, such as cyberbullying [7,8,9]. The current research intends to incorporate an important individual cognitive factor as an agent for familiar factors to exert its influence on cyberbullying.

### 1.1. Relationship between Parent–Child Attachment and Cyberbullying

Parental factors are important contributing factors in the microsystem and they have a significant effect on cyberbullying [8]. “A controlling parenting style as well as an inconsistent internet-mediation style were associated with a higher prevalence of adolescent involvement in cyberbullying as victims and as perpetrators” [10]. Moral disengagement is a self-regulation process that occurs when individuals are in need of reducing tension caused by their behaviors (such as bullying) not matching their moral standards [11]. A high level of moral disengagement of adolescents’ fathers leads to inappropriate childhood parenting, which positively predicts their moral disengagement, and in turn positively foretells cyberbullying [11]. In contrast, a better parent–child relationship is associated with a lower likelihood of adolescent cyberbullying [8].

Parent–child attachment (PCA) is an essential medium of parental influence on adolescents. PCA refers to the long-lasting, sustained emotional connection between a child and their caregiver [12]. Individuals internalize and integrate early childhood PCA interaction into their internal working model, which in turn, impacts their subsequent social adaptation proportionally. For example, secure attachment leads to individuals’ good social adaptation [13,14], whereas other types of attachment can cause maladaptation. Additionally, PCA is negatively associated with cyberbullying in high and middle school student groups [15,16], and parental emotional warmth is also significantly and negatively related to cyberbullying [17].

### 1.2. Relationship between PCA, Belief in a Just World and Cyberbullying

Personal cognitive styles and beliefs likewise influence individuals’ levels of aggressive behavior [18,19]. The belief in a just world (BJW) refers to people’s belief that they live in a just and orderly world where they will receive what they deserve and deserve what they receive [20,21]. Research has demonstrated the important link between BJW and cyberbullying behaviors. For instance, it was found that improving a personal BJW can reduce cyberbullying behaviors [7,22,23]. Furthermore, individuals with good PCA show more pro-social behaviors [13]. These research studies essentially provided evidence for the direct link between PCA and BJW, and between BJW and cyberbullying.

In addition to the direct relation between PCA and BJW, and between BJW and cyberbullying, it was found that the effect of psychological abuse in childhood on cyberbullying among college students can be reduced through the mediating role of BJW [24]. Together, these findings suggest that after internalizing good parent–child interactions, individuals would have more optimistic expectations of the outside world and more pro-social behaviors, thus reducing the emergence of aggressive behaviors in cyberspace.

### 1.3. The Role of School Climate

According to the ecological systems theory [6], in addition to parents, schools play an important role in adolescents’ life and it is an important factor in the microsystem. School climate refers to the quality of interpersonal interaction and opportunities for autonomy experienced or perceived by students at school [25]. A supportive school climate is associated with more positive cognition and behaviors of students [9,26]. The better the school climate perceived by middle school students, the lower the likelihood of their developing bullying behaviors [9]. With a higher level of school climate, individuals are more likely to defend the victim in cyberbullying circumstances [26]. In addition, a good school climate can improve adolescents’ BJW [27]. 

Based on the ecological systems theory, [28] proposed the systems view of school climate to further refine the conceptualization of school climate. From this theory, not only can various forms of support provided by the school climate serve as protective factors against cyberbullying among middle school students, but equally important, the nanosystem that exists within the school, which is a type of microsystem, also plays a role that cannot be ignored. Systems view theory suggests that in addition to the interaction between microsystems, such as schools and families, which constitute a mesosystem, there is also interaction between nanosystems, such as classes and peer groups, and microsystems, such as schools and families. These interactions together influence the establishment and quality of school climate. Schools as microsystems are small enough units, but due to classroom differences and individual trait level differences, the level of school climate perceived by each individual would still be different. In summary, school climate plays a crucial and positive role in promoting and protecting adolescents in all their developmental tasks. Therefore, for this study, we expect that school climate may also play a protective role in the relationship between PCA and individual cyberbullying through BJW.

### 1.4. Gender and Cyberbullying

According to the hostile attribution bias theory, in ambiguous situations, aggressive individuals are more likely to misinterpret the other party’s intention in the situation, attributing others’ motives toward a hostile direction, and thus engage in aggressive actions [29]. In turn, the aggressive behaviors lead others to take reactive aggressive behaviors, thus creating a vicious cycle [30,31]. Ref. [32] found that the association between physical aggression and the propensity for hostile attribution bias was stronger in males. The focus of our current study does not include gender difference on cyberbullying behaviors; however, we believe it is an important aspect that continues to draw researchers’ attention. Therefore, we will not include it into our research hypotheses, but will still compare gender difference on cyberbullying in the current sample. 

### 1.5. Research Hypotheses

Based on the ecological system theory, the possible important mediating role of individual cognitive factors (BJW) between family factors (PCA) and behavioral variables (cyberbullying behaviors) is important and has yet been investigated. The main purpose of the current study is to investigate the mediating role that BJW might play in the relationship between PCA and Cyberbullying. By so doing, we will be able to fill the research gap through understanding the interrelation between family factors (PCA) and individual behavior variables (cyberbullying behaviors) via the pathway of individual cognition (BJW).

The Hypothesis 1 (H1) of the current study is that PCA is positively correlated with BJW. The Hypothesis 2 (H2) is that BJW is negatively related to cyberbullying. Our Hypothesis 3 (H3) is that BJW mediates the relationship between PCA and cyberbullying. In addition, there is a lack of systematic exploration of possible intrinsic relationships between family factors (PCA), individual cognitive factors (BJW), school factors (school climate) and behavioral variables (cyberbullying). Hence, the Hypothesis 4 (H4) is that school climate moderates the relationship between PCA, BJW and Cyberbullying. This study contributes to the literature by systematically examining the combined effects of individual, family and school factors on cyberbullying behaviors. The hypotheses are depicted in Figure 1.

## 2. Methods

### 2.1. Participants

A total of 750 students from four middle schools in Dongguan and Shenzhen, Guangdong province, China were recruited for the study during school year in 2019. With a return rate of 96.8%, 726 questionnaires were collected and a final 706 valid questionnaires were obtained after rejecting invalid questionnaires. Among the participants, 369 were males, 337 were females and their age was between 10 to 16 years, with a mean age of 12.9 years (*SD* = 0.98).

### 2.2. Instruments

#### 2.2.1. The Inventory of Parent and Peer Attachment

The Inventory of Parent and Peer Attachment (IPPA), developed by [33], comprises three subscales for father, mother and peer, covering three dimensions of trust, communication and alienation. In this study, the short PCA version of the Parent and Peer Attachment Questionnaire was used [13] to measure PCA. The scale consists of 13 items and is scored directly on a 5-point Likert scale, ranging from *1 = Never* to *5 = Always*. A high score indicates greater security in PCA. A sample item is “My parents respect my feelings”. The Cronbach’s alpha coefficients for both the father and mother part were 0.93 [13]. The internal reliability of the IPPA scale in this study was 0.91.

#### 2.2.2. Cyberbullying Inventory

The Cyberbullying Inventory developed by [34] contains two subscales for cyberbullying and cyber victimization. The cyberbullying subscale, with 18 items, was used in this study to measure cyberbullying. The 4-point Likert scale was used to score the frequency of cyberbullying directly, ranging from *1 = Never* to *4 = More than five times*, with high scores indicating individuals’ high frequency of cyberbullying. A sample item is “To hurt someone you know through internet”. The internal reliability of the cyberbullying inventory was 0.83.

#### 2.2.3. Belief in a Just World Scale (BJWS)

We used the BJWS developed by [35] to measure BJW. The scale consists of personal BJW (7 items) and general BJW (6 items). All the items are scored directly on a 6-point Likert scale, from *1 = Disagree very strongly* to *6 = Agree very strongly*, with high scores indicating strong BJW. A sample item is “I think the world is mostly just”. In this study, the internal reliability of the BJWS was 0.88.

#### 2.2.4. School Climate Scale

A 25-item questionnaire on adolescents’ perceived school climate, revised by [36], was used to measure school climate. It consists of three dimensions of school climate: teacher support, student-student support and opportunities for autonomy. The items are scored directly on a 4-point Likert scale, ranging from *1 = Never* to *4 = Always.* A high score indicates a more positive school climate as perceived by adolescents. The validity of the three subscales was 0.81 and 0.84; 0.86 and 0.82; and 0.69 and 0.70, respectively [36]. A sample item is “My schoolmates are mean to each other”. In this study, the internal consistency coefficient of the School Climate Scale was 0.86.

### 2.3. Procedures

The study was approved by the Ethics Committee of Guangzhou University (protocol code GZHU2019007 and 27 May 2019) for studies involving humans. Before administering the questionnaires, permission was obtained from each school leader and informed consent was obtained from each middle school student participant. The psychology research assistants at each school administered the questionnaires. The research assistants were trained on the procedures before administering the questionnaires. We emphasized in the instruction the importance of completing the questionnaires independently and anonymously, and that there was no right or wrong answer. Participants were simply required to fill in answers that were suitable for them and they could choose to withdraw from the research any time during the process. The questionnaires were paper-pencil form and were administered in classes within 15–20 min. The questionnaires were then collected on-site. Some of the items in the questionnaire were reverse scored, and anonymity and confidentiality were highlighted in the instructions to reduce participants’ concern.

### 2.4. Data Analysis

In this study, we used SPSS 23.0 to conduct descriptive statistical analysis of each variable. Since psychological variables are often considered latent variables [37], it is more appropriate that we utilize software that deems variables as latent. Therefore, we used Mplus 7.4 to examine the correlation among all latent variables and the mediating effects of BJW. Lastly, we used Model 59 of [38] PROCESS for SPSS to test the moderating effect of school climate on the relationships between PCA, BJW and Cyberbullying. We chose to use SPSS to perform moderation analysis that we needed to perform simple slope tests that are available in SPSS. In the analysis process, we also controlled for variables, such as gender and age. In this paper, we assumed the missing data are MCAR [39].

## 3. Results 

### 3.1. Common Method Bias

To reduce common method bias due to only using questionnaires for data collection, both procedural controls and statistical tests were used in this study. Regarding the statistical tests, Harman’s one-factor test shows that there are 15 factors with an eigenvalue of greater than 1, and the contributing rate of the first common factor is 18.56%, which is much lower than 40%, indicating that the effect of the common method bias could be ignored in this study.

### 3.2. Descriptive Statistics and Correlation among Variables

The results showed that 47.5% of the students conducted cyberbullying behaviors against others in the past semester (an endorsement of 2 and/or above). Table 1 shows the differences in mean scores and standard deviations between the male and female groups for each variable. Cyberbullying is significantly higher among males than females (*t* = 3.56; *p* = 0.001), whereas perceived school climate is significantly higher among females than males (*t* = −3.29, *p* = 0.001). The mean of PCA and BJW does not differ significantly by gender.

The correlations among the variables showed that age was significantly and positively correlated with cyberbullying and was significantly and negatively correlated with PCA and school climate. Cyberbullying was significantly and negatively correlated with PCA, BJW and school climate. PCA was significantly and positively correlated with BJW and school climate, whereas BJW was significantly and positively correlated with school climate. Please refer to Table 2 for detailed information. 

### 3.3. Testing the Mediating Role of BJW and the Moderating Role of School Climate

After controlling for gender and age, we examined the relationship between PCA and cyberbullying through regression analysis. The results showed that PCA was significantly correlated with cyberbullying (*b* = −0.24, *SE* = 0.04, *p* < 0.001). The mediating role of BJW in the relationship between PCA and cyberbullying was also tested. Without incorporating BJW, PCA was significantly and negatively correlated with cyberbullying (*b* = −0.24, *SE* = 0.04, *p* < 0.001). When the mediating variable BJW was included (as shown in Figure 2), the model fit well, *χ*^2^ = 2145.08, CFI/TLI = 0.92/0.92, and RMSEA = 0.04. However, after including BJW in the model, PCA was no longer significantly correlated with cyberbullying behaviors (*b* = 0.02, *SE* = 0.06, *p* = 0.75).

After identifying the mediating role of BJW, we tested whether school climate moderated the relationships between PCA and BJW, between BJW and Cyberbullying and between PCA and Cyberbullying (as shown in Table 3 and Table 4). The results indicated that the interaction term of PCA and school climate was not significantly correlated with BJW (*b* = −0.02, *p* = 0.76), the interaction term of BJW and school climate was significantly correlated with cyberbullying (*b* = 0.12, *p* < 0.001) and the interaction term of PCA and school climate was not significantly correlated with cyberbullying (*b* = −0.02, *p* = 0.29), suggesting that school climate moderated the relationship between BLW and Cyberbullying, which is the second half of the mediation model.

To explain the interaction effect of school climate and BJW on cyberbullying better, we also conducted a simple slope test in this study. For the test, high and low scores in school climate were chosen as one standard deviation above or below the mean score, respectively. The results for the effect of perceived school climate on cyberbullying are presented in Figure 3. BJW was negatively related to cyberbullying when school climate perceived by students was at a lower level (*b* = −0.10, *SE* = 0.01, *p* < 0.001). In contrast, when school climate was at a higher level, BJW was not significantly correlated with cyberbullying (*b* = 0.004, *SE* = 0.02, *p* = 0.77).

## 4. Discussion

### 4.1. Middle School Students’ Cyberbullying Behaviors

Facing negative phenomena in cyberspace, middle school students are often unable to deal with these phenomena rationally or reasonably due to their limited academic knowledge and life experience. This study showed that 47.5% of the students engaged in cyberbullying against others in the past semester, and the proportion was higher than 28%, as reported by [40]. In [40], they divided cyberbullying into 7 categories, and asked participants to report whether they have engaged in any of the behaviors in the past 6 months. The different measuring criteria could have contributed to the difference. Also, cyberbullying was more common among males than among females. Our current finding was in accordance with previous research, such that the relationship between physical aggression and the propensity for hostile attribution bias was stronger in males [30,31,32].

### 4.2. The Relationship between PCA, BJW and Cyberbullying

This study aimed to systematically explore the relationships between family factors (PCA), individual cognitive factors (BJW), school factors (school climate) and a problematic behavioral variable (cyberbullying). The Hypothesis 1 confirmed that PCA was positively correlated with BJW. This was understandable, for if one had formed a secure attachment style with their parents, they are more likely to experience the world as a safe one, hence taking the stance that one would earn what they deserve and vice versa [20]. The Hypothesis 2 was also confirmed and was consistent with previous finding. Specifically, one who held a belief in a just world is less likely to engage in cyberbullying [7,22,23]. 

Our Hypothesis 3 that BJW mediated in the relationship between PCA and cyberbullying has also been confirmed. Ecological systems theory [6] suggests that one’s family is a vital microenvironment that influences the development of individuals’ social adaptation. The first interpersonal relationship that individuals develop is with their parents, and this relationship lasts the longest. Attachment theory [12] argues that individuals develop a stable set of mental representations of themselves and relationships to others, or internal working models, early in their interactions with their caregivers. The models will continue to function throughout each person’s lifelong development, influencing their emotions and behaviors in all relationships. PCA is an important protective factor for internalizing and externalizing problematic behaviors [41,42]. Individuals who have good attachment with their parents develop more pro-social behaviors and are less likely to engage in cyberbullying. In addition, when individuals encounter situations in cyberspace that can lead to aggressive behaviors, it is also possible that they are also more willing to communicate with their parents and others based on good attachment and to seek help and more positive solutions in the process.

The higher the level of perceived parental warmth, the more just individuals would think the living environment is [27,43]. Conversely, the lack of parental care during adolescence reduces the level of personal BJW [44]. Warm parental support creates an amiable family atmosphere for the child, and parents who treat the child as an independent individual help increase the child’s trust in others and the outside world [43]. With a good PCA relationship as the foundation, children can experience the emotional warmth of their parents, trust them and develop an individual BJW.

According to ref. [45], reciprocal determinism views environmental factors, behaviors and individual factors as relatively independent, yet interacting and mutually determining theoretical entities. For an individual, their mental activities and behavioral patterns are influenced by their cognition, in addition to the interaction between the three factors. BJW measures people’s perceptions of justice and can serve as an essential mediating variable in transmitting environmental influences onto behaviors. When individuals have high levels of PCA and are fairly treated as independent entities, they are more likely to perceive their world and environment as safe and fair. The likelihood of interpreting others’ intentions as hostile in interpersonal situations is also reduced. Consequently, the possibility of cyberbullying is also lower.

### 4.3. The Moderating Role of School Climate

Hypothesis 4 of this study has been partially confirmed. A lower level of school climate is associated with the negative impact of a lower level of BJW on cyberbullying. In other words, individuals with low levels of BJW were more likely to engage in cyberbullying when the perceived school climate is not desirable. When the perceived school climate was at a higher level, the likelihood of cyberbullying was reduced, regardless of the individual’s level of BJW. It is possible that a better school climate can protect individuals with low levels of BJW from engaging in cyberbullying. According to the ecological systems theory [6], in addition to their parents, schools play an important role in adolescents life. According to [28], the systems view of school climate, in addition to the direct influence from nanosystems and microsystems, the interaction between nanosystems and microsystems could also influence individuals’ perceived level of school climate. It is highly likely that class and peer groups played an important role in this study. Middle school students are a group of students with a strong sense of community [46]. Adolescents were likely supported through peer-group interaction and classroom activities. This process effectively contributed to the improvement of school climate. When the level of BJW is low, a favorable school climate could exert its protective effect, hence conducive to a lower level of cyberbullying behaviors among these middle school students. School climate was not found to moderate the other two paths in the mediation model. 

### 4.4. Limitations and Implications

This study contributed to the literature by systematically understanding adolescents’ cyberbullying. However, this study still has some limitations. First, with a cross-sectional design, no causal inferences can be made about the relationships between variables. Researchers can further validate this mediating model through follow-up studies in the future. Second, the participants were recruited only from Guangdong province rather than from more diverse regions in China. Guangdong is a well-developed region comparing to other provinces. Therefore, the findings derived from this current sample cannot be deemed as representative of the characteristics of cyberbullying behaviors in China in general. In addition, [28]’s systems view theory of school climate emphasized that school climate is shaped and influenced by the differences in culture, policies and other macro systems in diverse regions. Economic and cultural differences between Guangdong province and other regions may also lead to different results. Therefore, future studies could expand the sample’s geographic representation and to increase the sample size to better testify the moderating effect of school climate on the relationship between BJW and cyberbullying. Such studies will be helpful in providing insight into the role of school climate plays in affecting cyberbullying among students from middle school and other types of schools. Additionally, it is important for future research to consider parent educational levels, domestic violence and other possible support adolescent could receive outside of their home, since these factors could influence adolescents’ behaviors directly or indirectly. 

Our study also has some practical implications. Firstly, schools could focus on promoting a more positive school climate from various perspectives in order to help students benefit from the protective effect of it when it comes to cyberbullying. One possibility is through counselors and teachers focusing more on promoting effective interpersonal skills. This way, adolescents could feel more supported at school and can more easily interact with their peers. Secondly, in familial environment, parents need to make effort in terms of cultivating a healthy parent–child attachment style. With this, adolescents could be more likely to interpret others as benign and safe. Finally, schools could focus on nanosystems, such that adolescents could help each other through monitoring behaviors and providing peer feedback in order to reduce cyberbullying behaviors.

## 5. Conclusions

(1)In conclusion, PCA was significantly and negatively correlated with cyberbullying after controlling for gender and age.(2)After the inclusion of BJW, PCA was no longer significantly associated with cyberbullying. That is, BJW significantly mediated the relationship between PCA and cyberbullying.(3)When incorporating school climate in the mediation model, school climate significantly moderated the second half of the mediation model from BJW to cyberbullying, thus providing empirical evidence for the protective effect of school climate on cyberbullying under low levels of BJW.

## Figures and Tables

**Figure 1 ijerph-19-06207-f001:**
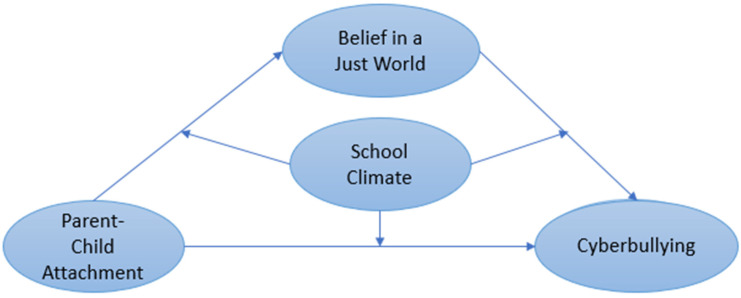
Hypothesized relationships.

**Figure 2 ijerph-19-06207-f002:**
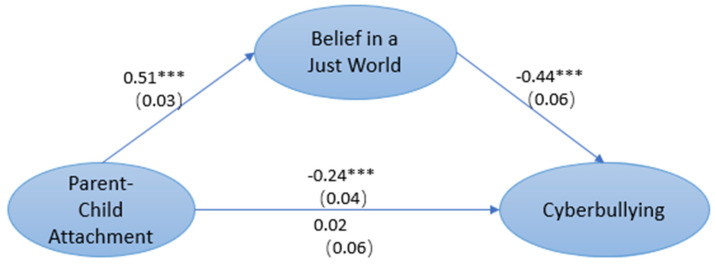
The mediation model between PCA, BJW and cyberbullying. Note: *** *p* < 0.001.

**Figure 3 ijerph-19-06207-f003:**
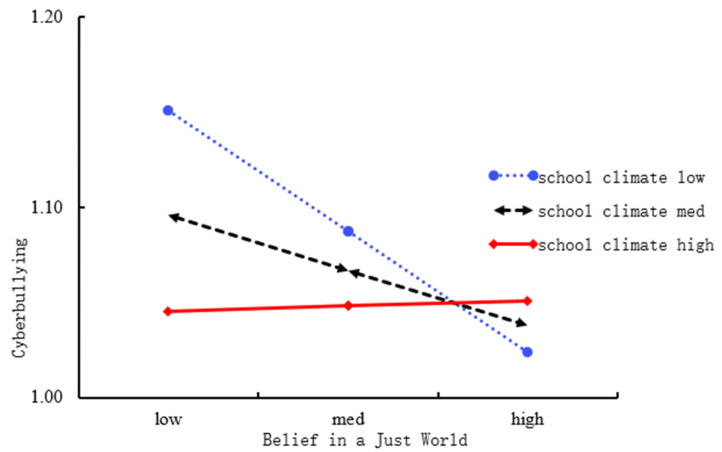
Simple slope for the interaction effect of belief in a just world and school climate on cyberbullying.

**Table 1 ijerph-19-06207-t001:** Gender differences in variables.

	Males (*n* = 369)		Females (*n* = 337)			
	*M*	*SD*	*M*	*SD*	*t*	Sig. (Two-Tailed)
Cyberbullying	1.11	0.24	1.06	0.1	3.56	0.001
PCA	3.56	0.74	3.53	0.92	0.43	0.668
BJW	3.32	0.69	3.35	0.7	−0.73	0.464
School climate	2.83	0.44	2.94	0.44	−3.29	0.001

**Table 2 ijerph-19-06207-t002:** Descriptive statistics and correlation coefficients of the variables.

Variables	*Mean*	*SD*	1	2	3	4	5	6
1. Dummy code gender	1	1	1					
2. Age	12.910	0.975	0.099 **	1				
3. Cyberbullying	1.083	0.191	0.129 **	0.185 **	1			
4. PCA	3.545	0.829	0.016	−0.192 **	−0.139 **	1		
5. BJW	3.334	0.698	−0.028	−0.066	−0.270 **	0.456 **	1	
6. School climate	2.883	0.445	−0.123 **	−0.170 **	−0.257 **	0.466 **	0.496 **	1

Note: ** *p* < 0.01, gender is a dummy variable, 1 = male, 0 = female.

**Table 3 ijerph-19-06207-t003:** Testing whether school climate moderated the relationship between PCA on BJW.

Result Variable	Predictors	*β*	*SE*	*p*
BJW	Gender	0.01	0.04	=0.83
	Age	0.04	0.02	=0.10
	PCA	0.30	0.17	=0.08
	School climate	0.64	0.21	=0.002
	PCA × School climate	−0.02	0.06	=0.76
	*R* ^2^	0.31	0.33	<0.001

**Table 4 ijerph-19-06207-t004:** Testing whether school climate moderated the relationship between PCA and cyberbullying and between BJW and cyberbullying.

Result Variable	Predictors	*β*	*SE*	*p*
Cyberbullying	Gender	0.04	0.01	=0.006
	Age	0.03	0.01	=0.0001
	PCA	0.07	0.06	=0.23
	School climate	−0.36	0.07	<0.001
	PCA × School climate	−0.02	0.02	0.29
	BJW	−0.38	0.07	<0.001
	BJW × School climate	0.12	0.02	<0.001
	*R* ^2^	0.19	0.08	0.03

## Data Availability

Not applicable.

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
