# Peer review of "The Relationship between Parent–Child Attachment, Belief in a Just World, School Climate and Cyberbullying: A Moderated Mediation"

_ijerph, 2022, doi:10.3390/ijerph19106207_

Round 1
Reviewer 1 Report
This study investigated the relationship between parental attachment and cyberbullying perpetrators, with mediators and moderators variables. A very few similar studies have been carried out among victims, but even fewer among perpetrators. Therefore, the knowledge provided is interesting and relevant.
One of my greatest concerns about this paper is the cited references. First of all, I am concerned that 15 of the citations are in the Chinese language, it will be very hard for most readers of this journal to understand the background. Second, I am concerned that some references are missing from the final list, for example, Martínez-Monteagudo, 2020 (line 27), which are cited in the text. Others are miscited, for example, Hinduja & Patchin, 2006, Patchin & Hinduja, 2006 is correct (line 395). I think it is important that the authors carefully review all the references.
I consider that the definition of cyberbullying is incomplete (line 24). Patchin & Hinduja define it as "willful and repeated harm inflicted through the medium of electronic text". In any case, I consider that there are some more adequate current definitions. Currently, harassment can occur without the need for text, for example through video.
I think the manuscript is scientifically sound and the experimental design is appropriate to test the hypothesis.
Manuscript results are reproducible based on the details given in the Methods section.
Figures and tables are appropriate and show properly the data.
Conclusions are consistent with the evidence and arguments presented
I think the ethics statements are adequate.
Author Response
Comments and Suggestions for Authors
This study investigated the relationship between parental attachment and cyberbullying perpetrators, with mediators and moderators variables. A very few similar studies have been carried out among victims, but even fewer among perpetrators. Therefore, the knowledge provided is interesting and relevant.
One of my greatest concerns about this paper is the cited references. First of all, I am concerned that 15 of the citations are in the Chinese language, it will be very hard for most readers of this journal to understand the background. Second, I am concerned that some references are missing from the final list, for example, Martínez-Monteagudo, 2020 (line 27), which are cited in the text. Others are miscited, for example, Hinduja & Patchin, 2006, Patchin & Hinduja, 2006 is correct (line 395). I think it is important that the authors carefully review all the references.
We thank the reviewer’s comments. We have now carefully proofread the citations in text and the reference list, and we added those that were missing. Regarding the citations in Chinese, we appreciate the reviewer’s careful read, and yes there are 15 Chinese papers out of 46 references in total that we cited. We would like to say that this paper is regarding Chinese adolescents’ cyberaggression, and we believe that it is inevitable that we focus on Chinese scholars’ work regarding this area. We completely agree with the reviewer’s concern that it might impose some difficulties for the readers, and we will take it into consideration in our future work, and we will try to incorporate more Chinese scholars’ work that is published in English journals.
I consider that the definition of cyberbullying is incomplete (line 24). Patchin & Hinduja define it as "willful and repeated harm inflicted through the medium of electronic text". In any case, I consider that there are some more adequate current definitions. Currently, harassment can occur without the need for text, for example through video.
Thank you very much for the thoughtful note. We now have incorporated your recommendation to make the definition more complete.
I think the manuscript is scientifically sound and the experimental design is appropriate to test the hypothesis.
We thank the reviewer’s confirmation. This means a lot to us.
Manuscript results are reproducible based on the details given in the Methods section.
We thank the reviewer’s confirmation.
Figures and tables are appropriate and show properly the data.
Thanks to the reviewer’s confirmation. We appreciate it.
Conclusions are consistent with the evidence and arguments presented.
We thank the reviewer’s confirmation. This means a lot to us.
I think the ethics statements are adequate.
Thanks to the reviewer’s confirmation.
Reviewer 2 Report
Overall, I thought the article was well-written but there are sections that need great work. The authors need to consistently stay on track and make sure all sections are in line with their hypotheses. The issues are stated below:
1) The authors need to define certain terms that are not as easily understood to someone who is not as familiar with the literature. For instance, just world belief in the abstract and moral disengagement discussed on pages 2-3. I recommend the authors go through and check for other terms that are a bit obscure.
2) The authors need to highlight how their study is filling a gap in the literature. It should be easily understood why their study is worth publishing and how it fills a gap in the current literature.
3) I recommend the authors make a separate subtitle for 'hypotheses' so it's separate from the gender section.
4) The gender section needs to be completed updated. There is no discussion of gender and cyberbullying in this section except for one sentence. There is a lot of literature on gender and cyberbullying yet none is discussed in this section. It does not make sense. On that same note, the authors, again, need to describe any theories discussed and not assume the reader is familiar with them.
5) I recommend the authors take out any sections in the literature review that are not directly related to their hypotheses and study. In addition, they need to greatly expand on the sections that do relate directly. Each of the variables should be discussed in great detail. A few sentences per section won't cut it.
6) For the statistical techniques applied in the study, a couple sentences should be included stating why these certain techniques were used and deamed the most applicable for this particular research.
7) Need a Variable Section. Every variable that was used in the study should be operationalized in section in the Methods. The reader shouldn't have to guess how a term is operationalized.
8) Be clear and cohesive. If you're not applying certain theories or variables in your study, then they shouldn't be discussed in the paper. Refer back consistently to the hypotheses. Stay on track.
Author Response
Comments and Suggestions for Authors
Overall, I thought the article was well-written but there are sections that need great work. The authors need to consistently stay on track and make sure all sections are in line with their hypotheses. The issues are stated below:
1) The authors need to define certain terms that are not as easily understood to someone who is not as familiar with the literature. For instance, just world belief in the abstract and moral disengagement discussed on pages 2-3. I recommend the authors go through and check for other terms that are a bit obscure.
We thank the reviewer’s comments, and we agree that it is important to provide a clear definition of important concepts that are mentioned in the text. We added the definition of moral disengagement in the text and highlighted the definition of belief in a just world in the text. We also made sure that other important variables in the paper have necessary definitions. Thank you!
2) The authors need to highlight how their study is filling a gap in the literature. It should be easily understood why their study is worth publishing and how it fills a gap in the current literature.
We thank the reviewer’s comments and we agree with the reviewer that it is essential that we provide the readers with an understanding of the contribution that our manuscript could make to the literature. Therefore, we added the relevant information in section 1.4.
3) I recommend the authors make a separate subtitle for 'hypotheses' so it's separate from the gender section.
Thanks to the reviewer’s comment regarding adding a separate section for the research hypotheses. We think it is logical to do so, and we have added an additional section in the text.
4) The gender section needs to be completed updated. There is no discussion of gender and cyberbullying in this section except for one sentence. There is a lot of literature on gender and cyberbullying yet none is discussed in this section. It does not make sense. On that same note, the authors, again, need to describe any theories discussed and not assume the reader is familiar with them.
We thank the reviewer’s comments and completely agree with it. We have carefully read our manuscript and realized that gender difference on cyberbullying is not our focus in the current study. However, it is of great importance, and we do not intend to ignore it and omit it from our paper. Therefore, we added some statements in the text to explain our stance on this: it is not one of our research hypotheses, but we still analyzed the difference just so we do pay attention to this important variable. Consequently, the discussion section that touched on this part will be relatively short.
5) I recommend the authors take out any sections in the literature review that are not directly related to their hypotheses and study. In addition, they need to greatly expand on the sections that do relate directly. Each of the variables should be discussed in great detail. A few sentences per section won't cut it.
Thanks to the reviewer’s helpful comments and we completely agree with the reviewer’s understanding regarding the importance of relevance of the literature review. We have carefully edited our manuscript, and we have added relevant information to make the flow of the logic better. We believe that now the literature section provides a more coherent argument.
6) For the statistical techniques applied in the study, a couple sentences should be included stating why these certain techniques were used and deamed the most applicable for this particular research.
We thank the reviewer’s comments and we have added the statement regarding why we chose the procedures to perform the stated statistical analyses.
7) Need a Variable Section. Every variable that was used in the study should be operationalized in section in the Methods. The reader shouldn't have to guess how a term is operationalized.
We thank the reviewer’s comments and we completely agree that variables that are measured in any study needs to be clearly defined and operationalized. Under instruments section, we have provided introduction regarding the measures that we used to measure each variable of interest. We also highlighted the parts in terms of how we did so. We believe it is sufficient for the purpose of operationalizing the variables.
8) Be clear and cohesive. If you're not applying certain theories or variables in your study, then they shouldn't be discussed in the paper. Refer back consistently to the hypotheses. Stay on track.
Thanks to the reviewer’s helpful comments that we were able to review our paper again to catch any irrelevant details that could potentially derail the focus of our study. We have made sure that our 4 main hypotheses are the focus and we managed to focus on providing detailed discussion regarding them as well. Thank you again!
Reviewer 3 Report
Introduction
This gives a good overview of relevant research.
Methods
We need to know the date (months/year) the survey was carried out in. While important generally, this is especially so when studying internet related phenomena. This should also be contextualized relative to the covid pandemic – was this survey pre-pandemic?
Results
Line 180 ’47.5% of the students conducted cyberbullying behaviors’ – this is very imprecise given we have 18 items on a 4-point scale. Also nearly half is a very high % of cyberbullies so presumably a lenient criterion was used here. This needs to be explained more fully.
Discussion
Lines 234-236 again the 47.5% needs to be contextualised – the % depends on the measurement criterion. Did Rao et al. 2017 use the same scale/criterion? The Rao reference is missing so needs to be added.
In 4.3, the ‘nanosystem’ is mentioned a couple of times. So far as I am aware, this is not a term used in Bronfenbrenner’s ecological model. So IF it is important for Discussion it should be explained a bit more, perhaps also in the Introduction.
In 4,4, it is correctly stated that ‘no causal inferences can be made’ from a cross-sectional study. In that light, the wording on line 287 earlier ‘can increase’ might be better as ‘is associated with’. Also, the results are consistent with the model in Figure 2, but other models could not be ruled out, and this could be stated.
Author Response
Comments and Suggestions for Authors
Introduction
This gives a good overview of relevant research.
Thanks to the reviewer’s confirmation. We appreciate it.
Methods
We need to know the date (months/year) the survey was carried out in. While important generally, this is especially so when studying internet related phenomena. This should also be contextualized relative to the covid pandemic – was this survey pre-pandemic?
Thanks to the reviewer’s comments. We completely agree with the reviewer’s understanding. We have now added the information.
Results
Line 180 ’47.5% of the students conducted cyberbullying behaviors’ – this is very imprecise given we have 18 items on a 4-point scale. Also nearly half is a very high % of cyberbullies so presumably a lenient criterion was used here. This needs to be explained more fully.
Thanks to the reviewer’s comment. We completely agree with the reviewer’s understanding that it is a lenient cutoff score for cyberbullying behaviors in the study. We did consider an endorsement of 2 and/or above as an indicator of engaging in cyberbullying behaviors, and we have added it in the text. We have now also added more to the discussion section so that logically it makes more sense to the readers.
Discussion
Lines 234-236 again the 47.5% needs to be contextualised – the % depends on the measurement criterion. Did Rao et al. 2017 use the same scale/criterion? The Rao reference is missing so needs to be added.
Thanks to the reviewer’s comment. We have now added the reference. In Rao’s paper, they did not use the measurement that we used in this study. They divided cyberbullying into 7 categories, and asked participants to report whether they have engaged in any of the behaviors in the past 6 months. We have added this difference in the text to explain the different results.
In 4.3, the ‘nanosystem’ is mentioned a couple of times. So far as I am aware, this is not a term used in Bronfenbrenner’s ecological model. So IF it is important for Discussion it should be explained a bit more, perhaps also in the Introduction.
We would like to thank the reviewer for the comments. We completely agree with the reviewer’s suggestion that it makes the logic of the study flow better if we were to add the systems view of school climate in the introduction part. Hence, we did so.
In 4,4, it is correctly stated that ‘no causal inferences can be made’ from a cross-sectional study. In that light, the wording on line 287 earlier ‘can increase’ might be better as ‘is associated with’. Also, the results are consistent with the model in Figure 2, but other models could not be ruled out, and this could be stated.
We would like to thank the reviewer for the comments. We have corrected the wording in the text. We have also added the statement that the other two paths were not confirmed by the results.
Reviewer 4 Report
The article ' Relationship between Parent-Child Attachment, Belief in a Just World, School Climate and Cyberbullying: A Moderated Mediation ', presented for review is interesting. The research methods used are well selected. This study contributed to the literature by systematically understanding adolescents’ cyberbullying. However, some issues need to be clarified or supplemented. The comments are included below.
Title
- The title is worded correctly and accurately reflects the content.
Abstract
- The abstract is clear and adequate.
- Introduction
- The aim of the study is well defined but not very visible. The aim of the study can be presented in a separate paragraph.
-In future research or literature review it is worth considering:
* level of education in the family,
* providing the child with the possibility of relieving stress
* symptoms of domestic violence
These are difficult topics to verify, but they have a great influence on the child's behavior.The possible important mediating role of individual cognitive factors between family factors and behavioral variables is important and not has yet been very well investigated.
- Table 2. - Please verify the correctness of the numerical values in the table.
- Methods
- Correct research methodology.
- Results
- A clear description of the results, supported by a discussion with the literature.
- Discussion
The discussion of the results was carried out meticulously and exhaustively.
- Conclusions
- Conclusions is appropriate.
Final conclusion - It is worth carrying out larger research in a larger geographic area in the future.. Such studies will be helpful in providing insight into the role of school climate plays in affecting cyberbullying among in the public school students.
Author Response
Comments and Suggestions for Authors
The article ' Relationship between Parent-Child Attachment, Belief in a Just World, School Climate and Cyberbullying: A Moderated Mediation ', presented for review is interesting. The research methods used are well selected. This study contributed to the literature by systematically understanding adolescents’ cyberbullying. However, some issues need to be clarified or supplemented. The comments are included below.
Title
- The title is worded correctly and accurately reflects the content.
We thank the reviewer’s confirmation.
Abstract
- The abstract is clear and adequate.
Thanks to the reviewer’s confirmation. This means a lot to us.
- Introduction
- The aim of the study is well defined but not very visible. The aim of the study can be presented in a separate paragraph.
Thanks to the reviewer’s comments. We have now started a new paragraph for the purpose of the study and added a separate section for our research hypotheses.
-In future research or literature review it is worth considering:
* level of education in the family,
* providing the child with the possibility of relieving stress
* symptoms of domestic violence
These are difficult topics to verify, but they have a great influence on the child's behavior.The possible important mediating role of individual cognitive factors between family factors and behavioral variables is important and not has yet been very well investigated.
Thanks to the reviewer’s recommendation. We have now included this important point in the limitation and implication section.
- Table 2. - Please verify the correctness of the numerical values in the table.
Thanks to the reviewer’s reminder and we have carefully proofread the accuracy of the numbers presented in the table.
- Methods
- Correct research methodology.
Thanks to the reviewer’s confirmation. We appreciate it.
Results
- A clear description of the results, supported by a discussion with the literature.
We thank the reviewer’s confirmation. This means a lot to us.
- Discussion
The discussion of the results was carried out meticulously and exhaustively.
Thanks to the reviewer’s confirmation. We appreciate it a lot!
- Conclusions
- Conclusions is appropriate.
We thank the reviewer’s confirmation. This means a lot to us.
Final conclusion - It is worth carrying out larger research in a larger geographic area in the future.. Such studies will be helpful in providing insight into the role of school climate plays in affecting cyberbullying among in the public school students.
We thank the reviewer’s comment. We have added more information to the implication to incorporate the reviewer’s suggestion.
Round 2
Reviewer 1 Report
The errors/irregularities reported have been dealt with by the authors.
Reviewer 2 Report
Overall, Looks good